# A Posture Recognition Method Based on Indoor Positioning Technology

**DOI:** 10.3390/s19061464

**Published:** 2019-03-26

**Authors:** Xiaoping Huang, Fei Wang, Jian Zhang, Zelin Hu, Jian Jin

**Affiliations:** 1Institute of Intelligent Machines, Chinese Academy of Sciences, University of Science and Technology of China, Hefei 230031, China; hxping@mail.ustc.edu.cn (X.H.); kljjab@163.com (J.J.); 2School of Electronics and Information Engineering, Anhui University, Hefei 230039, China; jswangfei1019@163.com; 3Institute of Intelligent Machines, Chinese Academy of Sciences, Hefei 230031, China; zlhu@iim.ac.cn

**Keywords:** posture recognition, indoor positioning, wireless body area network, Kalman filtering, multi-sensor combination

## Abstract

Posture recognition has been widely applied in fields such as physical training, environmental awareness, human-computer-interaction, surveillance system and elderly health care. The traditional methods consist of two main variations: machine vision methods and acceleration sensor methods. The former has the disadvantages of privacy invasion, high cost and complex implementation processes, while the latter has low recognition rate for still postures. A new body posture recognition scheme based on indoor positioning technology is presented in this paper. A single deployed indoor positioning system is constructed by installing wearable receiving tags at key points of the human body. The distance measurement method with ultra-wide band (UWB) radio is applied to position the key points of human body. Posture recognition is implemented by positioning. In the posture recognition algorithm, least square estimation (LSE) method and the improved extended Kalman filtering (iEKF) algorithm are respectively adopted to suppress the noise of the distances measurement and to improve the accuracy of positioning and recognition. The comparison of simulation results with the two methods shows that the improved extended Kalman filtering algorithm is more effective in error performance.

## 1. Introduction

Human posture recognition is an attractive and challenging topic due to its wide range of applications, e.g., smart home environments for the monitoring of physical activity levels, assessment of recovery phases of living independently, and detection of accidental falls in elderly people [1]. Among these applications, the most important one is the elderly health care due to the population aging in the 21st century. According to the US population report, the aged population (over 65 years old) reached more than 50 million in 2017 [2], which represented 15.41% of the US population. China had an aged population of 158 million (10.64% of its total population) in 2017, and it will become one of the most aging countries in the world [3]. Meanwhile, the empty nest ratio of the elderly is rapidly increasing for various reasons. Therefore, health care for the elderly has become a major concern. Posture recognition is one of the key supporting technologies for health care of the elderly.

Traditional human posture detection methods can mainly be divided into two categories: computer vision [3,4,5,6,7,8,9,10,11,12] and acceleration sensor data analysis [13,14,15,16]. Methods based on acceleration sensors have the disadvantage of complex data processing steps, but their invasion of privacy is well tolerated. Posture detection methods by computer vision technology are mature and have high accuracy in individual posture recognition, but they have the disadvantage of invasion of privacy. Posture parameters extracted from video in [4,5] are skeletal joints and rotation angles. The method performs well in individual human posture recognition with a pan-and-tilt fixed camera, but it suffers difficulties in the recognition of multiple people. Kinect, the Microsoft somatosensory camera, is adopted to extract such parameters as spatial positions of the skeletal joints [6,7,8,9,10,11] to recognize human posture. Kinect can track at most two bones, six people, 20 joint points in a standing model, or 10 joint points in a sitting model [12]. Due to its poor recognition effect under conditions of multiple participants, Kinect doesn’t meet the requirement of multiple people tracking. Both cameras and Kinect have the disadvantage of invasion of privacy, which result in controversies in health monitoring application for the elderly. 

On the other hand, human posture detection technology based on acceleration sensor data analysis has been proposed in [13,14,15,16]. In [13], a waist-mounted device to detect possible falls of elderly people is presented, and four accelerometer sensors are combined to achieve good performances. However, the algorithm needs to be improved to calculate the optimum thresholds automatically. Musalek [14] used a motion sensor, which is a wearable device capable of wireless communication, to detect the movement of the elderly. Both methods depend on single sensor devices, which provides limited information to recognize human posture. Guo et al. [15] proposed a pose awareness solution for estimating pedestrian walking speeds with the sensors built into smartphones. This method asks the elderly to use a high cost smartphone. Reference [16] presents wearable sensor devices to recognize human postures, and they conducts their experiments with participants wearing three sensors which can reach 90% overall accuracy of human postures. Caroppo et al. [17] described a multi-sensor platform for anomalies, which acquires postures by both ambient and wearable sensors that are a time-of-flight 3D vision sensor, UWB radar sensor and a 3-axis accelerometer. The platform achieves high accuracy in sleep anomaly detection. 

Methods based on acceleration sensors can achieve high recognition accuracy in motion states. Since acceleration sensors cannot acquire static information, they have difficulties in the recognition of positions, shapes, and so on. In recent years, the indoor positioning systems have seen increasing development [18,19,20,21,22,23,24]. The Microsoft indoor positioning competition has attracted a large number of global teams from both companies and universities each year. The competition has witnessed lots of positioning technologies including wireless local area networks (WLAN) [19], Bluetooth low energy [20], optical light [21], radio frequency identification (RFID) [22], and UWB [23,24]. Among these methods, UWB is considered to be one of the most accurate approaches because it provides positioning estimation with centimeter-level accuracy [25,26]. 

However, it is doubtful whether the indoor position system can also offer high-accuracy position estimation and posture recognition in dynamic activities. To answer this question, a new posture recognition method with the application of UWB indoor positioning technology is proposed in this paper. In the scheme, a positioning umbrella is designed and constructed first, the UWB distance measuring technology is applied. Receiving tags will be pasted onto key points on the clothes corresponding to human body joints of such as wrists and ankles. Least squares and Kalman filtering algorithms are adopted to reduce the noise interference, thus further improving the positioning and recognition accuracy. Simulation results reveal that the scheme can effectively estimate positions and recognize human postures.

## 2. Posture Estimation Method

### 2.1. Design of Positioning System

UWB is one of the hotspots in indoor positioning research. It can achieve centimeter-level accuracy in positioning, and has good multipath resistance performance. The radio can transmit a long distance with low power consumption. Hence, the hardware of the positioning system proposed in the paper is designed based on UWB. The positioning system consists of two parts: a positioning umbrella and receiving tags, as is shown in Figure 1. To facilitate attachment on clothes, the receiving tags are designed to be miniature and hidden. The positioning umbrella is the key component of positioning system. It is made up of core and arms, as shown in Figure 1. The core of the positioning umbrella is composed of a CPU and other control circuit modules such as a communication module, alarm executor module, and location estimation module. The arm of the positioning umbrella connects the UWB radio sender to the central processing unit (CPU), so the senders can work synchronously under the control of the CPU. The UWB module mainly functions as a time of flight (TOF) device for distance measurement. 

There are several arms in a positioning umbrella system. However, positioning in two-dimensional surface needs at least three arms, and positioning in three-dimensional space requires at least four non-coplanar arms. The more and longer the arms are, the higher the positioning accuracy is [27]. Due to space limitations, the length of arms cannot be infinite. Arms in the system are limited to not more than 1 meter. Meanwhile, the number of arms is limited to eight. Thus, the design can be easy to install and use. As is shown in Figure 1a, the distribution of seven positioning umbrella arms in space can provide position service for any UWB receiving tags in 3-dimensional space.

Powered by a button battery, the receiving tags attached to coats and caps are designed to be very small. In order to ensure low energy consumption, the positioning algorithm should be concise and effective. Positioning algorithms with large computation burdens or long operation time (e.g., particle filters or unscented Kalman filter) do not meet the requirement. In order to make effective use of resources, tags attached to clothes, as is shown in Figure 1b, should be recovered and replaced.

### 2.2. Position Arrangement of Receiving Tags

The Microsoft Kinect technology collects more than 10 key joints of the human body to achieve posture recognition tasks. Similarly, we construct a human body model with 14 segments and 15 joints, as shown in Figure 2. The system applies 14 receiving tags attached to clothes (e.g., coats trousers and caps) in key joints of human body. By this method, we can easily detect human posture, for instance, we can determine it walking or standing by monitoring the position of receiving tags on knee joints. Also we can decide it falling down or sitting by detecting the position of head tag. For some complex postures such as picking up a cell phone, we need to analyze the combination of tags on hand joint, elbow joint and shoulder. We have named each receive tag and its corresponding position in Table 1. In the next two section (Section 2.3 and Section 2.4), the posture recognition method will be discussed based on these 14 receiving tags and their position arrangements.

### 2.3. Structure Vector of Human Body

Feature vectors and classifiers are always constructed to implement pattern recognition. Because receiving tags attached to clothes may differ in position for different persons, it is difficult to express the posture characteristics by the absolute position of receiving tags. Therefore, we introduce structure vectors to reproduce the postures of the human body. Due to the different characteristics of the human body, the structure vectors are constructed by the receiving tags which are attached to different positions of clothes to represent trunk, limbs and motion information. Since motion information is represented by a combination of receiving tags, behavior features can be obtained by calculating the vector modulus and vector angle. Compared to the method with one tag, the combination is a more effective method for posture recognition.

By analyzing the characteristics of human body, the vector by connecting receiving tags attached to the key joints is called *structure vector*. Suppose the position of the receiving tag attached to the left elbow is $A(x_{1},y_{1},z_{1})$, and position of the receiving tag on the left hand is $B(x_{2},y_{2},z_{2})$, the vector $\vec{AB}$ can be expressed as Equation (1):(1)$$\vec{AB}=(x_{2}-x_{1},y_{2}-y_{1},z_{2}-z_{1})$$

Other body structure vectors are similarly constructed. The combination of 14 key joints of Figure 2 in pairs results in 91 structure vectors, among which some vectors are useless to represent human postures. According to the structure characteristics of human body, the vector acquired by two adjacent joints contains the most abundant information. We choose 10 groups of structure vectors that are most capable of expressing changes in body posture, as shown in Table 2.

### 2.4. Vector Angle Setting

Besides structure vectors, angle relations between some vectors can effectively reflect the motion information. Figure 3 shows an example for the process of falling down. The angle of $I_{\mathrm{lhip}\text{-}\mathrm{to}\text{-}\mathrm{heart}}$ and $I_{\mathrm{lhip}\text{-}\mathrm{to}\text{-}\mathrm{lknee}}$ is dynamically changing, and the posture angle *θ* is changing synchronously. Meanwhile, angles can eliminate the structure vector differences originated from shapes and positions of various people.

The angle of the two structure vectors, $a=(x_{1},y_{1},z_{1})$ and $b=(x_{2},y_{2},z_{2})$, is defined as Equation (2):(2)$$\langle a,b\rangle =\arccos\frac{a\cdot b}{\left|a\right|\left|b\right|},\;\left|a\right|\neq 0\;\mathrm{and}\;\left|b\right|\neq 0$$

If |a|=0 or |b|=0, then 〈a,b〉=0, where a⋅b and |a| are respectively expressed as Equations (3) and (4):(3)$$a\cdot b=x_{1}x_{2}+y_{1}y_{2}+z_{1}z_{2}$$
(4)$$\left|a\right|=\sqrt{x_{1}^{2}+y_{1}^{2}+z_{1}^{2}}$$

Now, we make the naming rule of vector angles. The angle of vector $I_{relbow-to-rhand}$ and vector $I_{rshulder-to-relbow}$ is defined as $\theta_{rshoulder-relbow-rhand}$. According to the rule, we select 10 groups of vector angles containing best information that reflects the change of posture, as shown in Table 3.

## 3. Positioning Algorithm

### 3.1. Positioning Principles

According to TOF distance measurement principles of UWB, the angles and length of arms in positioning umbrella are known, and the UWB signal transmitting node at the endpoint of each umbrella arm functions as a base station (or an anchor node). The *i-*th (*i*∈{1,…*M*}) anchor has its position labeled as $P(x_{m}^{i},y_{m}^{i},z_{m}^{i})$. The anchor position is obtained once the umbrella is constructed. However, the positions of receiving tags are unknown. They will change with body motion. Since receiving tags move with human posture, their positions need to be estimated. Assuming that the position of the *j*-th (*j*∈{1,…*N*}) tag is $P(x_{n}^{j},y_{n}^{j},z_{n}^{j})$, the distance between the *j-*th tag and the *i*-th anchor node can be described in Equation (5), where *v* is the measurement noise which conforms to Gaussian distribution (v~N(0,R)), *R* represents variance of *v*, and $x_{n}^{j}$, $y_{n}^{j}$ and $z_{n}^{j}$ are three unknown position parameters. The resolution needs a set of equations with more than four equations similar to Equation (5), as shown in the equation set (6).
(5)$$Y_{i}^{j}=\sqrt{{\left(x_{m}^{i}-x_{n}^{j}\right)}^{2}+{\left(y_{m}^{i}-y_{n}^{j}\right)}^{2}+{\left(z_{m}^{i}-z_{n}^{j}\right)}^{2}}+v$$
(6)$$\{\begin{array}{c} Y_{1}^{j}=\sqrt{{\left(x_{m}^{1}-x_{n}^{j}\right)}^{2}+{\left(y_{m}^{1}-y_{n}^{j}\right)}^{2}+{\left(z_{m}^{1}-z_{n}^{j}\right)}^{2}}+v_{1}\\ Y_{2}^{j}=\sqrt{{\left(x_{m}^{2}-x_{n}^{j}\right)}^{2}+{\left(y_{m}^{2}-y_{n}^{j}\right)}^{2}+{\left(z_{m}^{2}-z_{n}^{j}\right)}^{2}}+v_{2}\\ \vdots \\ Y_{M}^{j}=\sqrt{{\left(x_{m}^{M}-x_{n}^{j}\right)}^{2}+{\left(y_{m}^{M}-y_{n}^{j}\right)}^{2}+{\left(z_{m}^{M}-z_{n}^{j}\right)}^{2}}+v_{M} \end{array}$$

Since there are seven arms in the positioning umbrella in Figure 2, the equation set (6) has seven equations. Linear processing of equation set (6) can be made before the three unknown parameters, and $x_{n}^{j}$, $y_{n}^{j}$ and $z_{n}^{j}$ are resolved by least squares estimation. Once the positions of receiving tags are solved, the structure vectors and angles can also be estimated.

### 3.2. Improved and Extended Kalman Filtering Algorithm

In Equation (5), UWB distance measurement is affected by noise *v*, which has great influence on positioning accuracy. Therefore, an effective algorithm to suppress the measurement noise must be introduced. Since the algorithm needs to be transplanted to a microprocessor, the data processing methods in the receiving tags can’t be too complicated for saving energy. Therefore, concise and effective methods such as least square algorithms and extended Kalman filtering algorithms are adopted in our application.

The position of each receiving tag is constructed as $X_{n}^{j}(k)=[x_{n}^{j},y_{n}^{j},z_{n}^{j}]$, which is called the state variable. Then state equation of these dynamic receiving tags can be represented as the following equation: (7)$$X_{n}^{j}(k+1)=\Phi X_{n}^{j}(k)+\Gamma W_{n}^{j}(k)\;\Phi =\left[\begin{array}{ccc} 1 & 0 & 0\\ 0 & 1 & 0\\ 0 & 0 & 1 \end{array}\right],\;\Gamma =\left[\begin{array}{c} 1\\ 1\\ 1 \end{array}\right]$$where Φ refers to state-driven matrix, Γ stands for noised-driven matrix, and *W* is white noise with mean value of 0 and the variance of *Q* (W~N(0,Q)). The observation equation is defined in the following form:(8)$$Y_{i}^{j}(k)=h(X_{n}^{j}(k))+v(k)$$where $h(X_{n}^{j}(k))=\sqrt{{\left(x_{m}^{i}-x_{n}^{j}(k)\right)}^{2}+{\left(y_{m}^{i}-y_{n}^{j}(k)\right)}^{2}+{\left(z_{m}^{i}-z_{n}^{j}(k)\right)}^{2}}$.

Then, extended Kalman filter is adopted to process the distance measurement noise. The detailed processing steps are described as follows:

(1) State Prediction:(9)$${\hat{X}}_{n}^{j}(k+1|k)=\Phi {\hat{X}}_{n}^{j}(k|k)$$

(2) Covariance Prediction:(10)$$P(k+1|k)=\Phi P(k|k)\Phi^{T}+Q(k+1)$$

(3) Kalman Gain Calculation:(11)$$K=P(k+1|k)H^{T}{\left[HP(k+1|k)H^{T}+R(k+1)\right]}^{-1}$$

The Jacobian matrix derived from Equation (11) is shown in Equation (12):(12)$$H=\frac{\partial h}{\partial X}=\left[\begin{array}{ccc} \frac{\partial h}{\partial x_{n}^{j}(k)} & \frac{\partial h}{\partial y_{n}^{j}(k)} & \frac{\partial h}{\partial z_{n}^{j}(k)} \end{array}\right]$$

(4) Status updating:(13)$${\hat{X}}_{n}^{j}(k+1|k+1)={\hat{X}}_{n}^{j}(k+1|k)+Ke\;e=(Y_{i}^{j}(k+1)-h({\hat{X}}_{n}^{j}(k+1|k)))$$

(5) Covariance updating:(14)P(k+1)=[I−KH]P(k+1|k)

The initial filtering value $X(0)=E\{\hat{X}(0)\}$, and the initial variance matrix $P(0)=\mathrm{var}\{\hat{X}(0)\}$. In the filter, *e* is the Kalman gain, which is calculated from historical data and the latest observation. Too much historical data will lead to cumulative errors. In order to avoid the accumulation of errors, many algorithms such as those described in [28,29] are improved through variable forgetting factors, but it is difficult for them to confirm the values. Therefore, we introduce the rectangular window function to improve the Kalman filter in this paper, as shown in the following form: (15)$$Y_{i}^{j}(k)=\{\begin{array}{c} f(Y_{i}^{j}(l)),\;N-k<l\leq k\\ 0,other \end{array}$$where $f(Y_{i}^{j}(l))=a_{0}+a_{1}x+\cdots +a_{k}x^{k}$, and *N* is the length of window function with its range 10≤N≤30. Another key improvement for the extended Kalman filter is that the polynomial fitting for the *N* latest observation $Y_{i}^{j}$ is employed. The fitting equation is shown in the following form:(16)$$A={\left(X^{T}X\right)}^{-1}X^{T}Y$$where $X=\left[\begin{array}{cccc} 1 & x_{1} & \cdots & x_{1}^{k}\\ 1 & x_{2} & \cdots & x_{2}^{k}\\ \vdots & \vdots & \ddots & \vdots \\ 1 & x_{n} & \cdots & x_{n}^{k} \end{array}\right]$, $A=\left[\begin{array}{c} a_{0}\\ a_{1}\\ \vdots \\ a_{k} \end{array}\right]$, $Y=\left[\begin{array}{c} y_{k-N+1}\\ y_{k-N}\\ \vdots \\ y_{k} \end{array}\right]$.

Equation (15) is easy to solve with the coefficient matrix *A* in Equation (16). Introduction of the window function in the improved algorithm aims at dropping historical data. The smoothed distance obtained by least square polynomial fitting with the latest *N* observation can help to reduce cumulative errors, and finally to improve the performance of Kalman filter.

## 4. Experiment

A sketch map of indoor environment is drawn in Figure 4a. The indoor playground is a square with a width of 10 m and length of 10 m. The positioning umbrella is suspended from the ceiling. It is suspended 5 m above the ground. There are seven arms with the length of 1 m. Among the seven arms, six are deployed in the *x*-*y* plane, and one is deployed in the *z*-direction. When the character moves under the umbrella, we capture the position of the receiving tag of the left foot (other parts are also possible) and draw the trajectory in Figure 4b, which shows the motion curve of the left foot when walking in a straight line at a constant speed. Human motion data for the simulation experiment is obtained from the Unreal Engine 4.0, which is a virtual reality software released in the USA. In this paper, we employ the engine to generate human posture data which are the positions of 14 tags when the character is walking. 

In the simulation experiment, the character walks several meters in a straight line at a constant speed. We set the sampling frequency as 119 fps, which means the positions of 14 receiving tags, such as head, shoulders, hands, knees, feet, and so on, should be recorded 119 times per second. Due to the high sampling frequency, we can see the details in the curve from Figure 4b. Then we employ our improved Kalman filtering (iEKF) algorithm. In Figure 4b, “Real” refers to data generated by Unreal Engine 4.0, “LSE” stands for results of least squares estimation (LSE) which is widely used in the field of indoor positioning technology, and “iEKF” represents the results of the improved extended Kalman filtering algorithm in Section 3.2.

## 5. Results Analysis and Discussion

While in Figure 4b, it is not obvious to decide which method is better. Both algorithms show good tracking effect in three-dimensional trajectories. For further comparison of the performances of two algorithms, we define deviation using Equation (17). Also the mean deviation of each algorithm is defined as Equation (18), where $X_{real}$ are the positions, structure vectors or angles, and $X_{estimate}$ are the estimations by iEKF and LSE methods. The following sections discuss the results of the experiment.
(17)$$deviation=\left|X_{estimate}-X_{real}\right|$$
(18)$$\mathrm{mean}-deviation=\frac{1}{N}\sum_{k=1}^{N}\left|X_{estimate}(k)-X_{real}(k)\right|$$

For further comparison, the mean deviation is an effective metric. The algorithm designed in Section 3 computes the positions of the tags, the vector norms and posture angles by applying iEKF or LSE method. We select two items, respectively, from Table 1, Table 2 and Table 3. The results of mean deviation are given in Table 4 for the LSE and iEKF methods, respectively. We can conclude that the iEKF is more effective because all the deviations are smaller than with the LSE method. 

(a) Positioning comparison

As described in Section 2.1, we choose 14 receiving tags attached on clothes corresponding to joints of the human body. The positions of the 14 key joints are easily obtained by our methods. In Figure 5, there are two groups of detection results. The first group are the 14 key joints which are detected by the indoor position system, and the others are detected by Microsoft Kinect technology [30]. 

The disadvantage of Kinect detection results is that they may be sheltered by something or some joints are missed. Moreover, Kinect cannot output the positions of the 14 key joints. However, these disadvantages are overcome by the indoor position system. The joints are missed only when receiving tags are out of power or broken in the proposal system. Unlike the Kinect method, the receiving tags can calculate the positions.

The positions of 14 joints of human body can be estimated by the iEKF and LSE methods. Taking the head tag for example, Figure 6 shows the deviation of estimation $X_{estimate}$ from real value $X_{real}$ obtained by the two algorithms for head tag. It reveals that iEKF has less deviation and better performance than the LSE method. The mean deviation by the iEKF method is 7.16 cm, lower than the LSE method one of 11.70 cm, so iEKF displays better position accuracy than the LSE method. 

(b) Struction vector comparison

We now perform further analysis to examine the deviation of structure vectors for body postures. Taking vector $I_{\mathrm{heart}\text{-}\mathrm{to}\text{-}\mathrm{lshoulder}}$ for example, in Figure 7, the red solid line represents the real value of distance between head and left shoulder, the green dotted line refers to the norm of vector $I_{\mathrm{heart}\text{-}\mathrm{to}\text{-}\mathrm{lshoulder}}$ estimated by the LSE method, and the blue dashed line represents the norm estimated by iEKF. Figure 7 shows that the norm of the vector fluctuates between 30.5 cm and 32.5 cm, which is due to the body tilting from left to right when the character is walking. The estimation values of both LSE and iEKF fluctuate near the true value. Figure 8 reveals that iEKF has a smaller deviation and is closer to the true value. The mean deviation between the real value and iEKF is 0.20 cm, which is lower than that of LSE method with 0.32 cm. Thus iEKF has better accuracy.

(c) Vector angle comparison

Vector angle reflects well the postures of the human body. Posture angles analysis play an important role in human posture recognition. Vector angle $\theta_{\mathrm{heart}\text{-}\mathrm{lshoulder}\text{-}\mathrm{lelbow}}$ is made up by vector $I_{\mathrm{lshoulder}\text{-}\mathrm{to}\text{-}\mathrm{lelbow}}$ and vector $I_{\mathrm{heart}\text{-}\mathrm{to}\text{-}\mathrm{lshoulder}}$. The angle usually happens under such a condition when people pick up a phone or take a glass of water to drink. In Figure 9, the red solid line, green dotted line and blue dashed line respectively represent the real angle, estimation by LSE and estimation by iEKF,. Also the estimation values of both LSE and iEKF fluctuate near the true value. However, Figure 10 indicates that the deviations of iEKF are lower than those of LSE. The mean deviation estimated by LSE is 0.0104 rad, and the deviation by iEKF is 0.0073 rad which is lower than the LSE method. The iEKF still has obvious advantages in the comparison of vector angles.

(d) Influence discussion for window length N

We refer to the improvement of iEKF in Section 3.2. Now we discuss the influence of the length of *N* for the window function. When *N* is set from 0 to 40, we carry out the experiments and calculate the mean deviation of vector $I_{\mathrm{lshoulder}\text{-}\mathrm{to}\text{-}\mathrm{lelbow}}$. The results are shown in Figure 11. From the figure, we can draw the conclusion that optimal performance is achieved when *N* is between 20 and 25. When *N* is equal to 23, the deviation is lowest. This doesn’t mean that it’s globally optimal when *N* is 23. The optimum *N* will float slightly for other vectors. Therefore, the ideal choice is 18≤N≤28 according to the simulation results.

(e) Future work

The outputs of the experiment will be two import products: one is the indoor position umbrella which can provide location services, and the other is the specific clothes with receiving tags which can recognize the postures of the human body. In the near future, we need to carry out the following three tasks before we can convert the experimental results into products:(1)We have done basic posture recognition work for a single person. In order to improve algorithm robustness, multi-person postures need to be tested.(2)The experiment is only at the stage of algorithm simulation. Nonetheless, our system hardware is completed, and testing work of transplanted algorithms will be done in the following stage.(3)The final target is to deliver this wearable device to the elderly. More and more postures will be tested such as walking, sitting, sleeping, crawling, calling, falling down, and so on. Evaluating the weaknesses of the entire system and optimizing tasks need to be tested many times.

## 6. Conclusions

The paper provided an overview of a posture recognition method for elderly care. The technologies available can be divided into two categories: vision-based recognition and sensor-based recognition. To avoid invasion of privacy, sensor-based recognition is a better choice. We proposed a sensor-based scheme for posture recognition with the indoor positioning system and receiving tags. The positioning umbrella with seven arms can provide location services. Meanwhile, the receiving tags are pasted on the surface of special clothes to measure the distance from umbrella by UWB radio. In this solution, we carried out simulation experiments to verify the usability of the scheme. The LSE method and iEKF algorithm are introduced to estimate the positions of receiving tags. We also present posture recognition algorithms with structure vectors and posture angles which combined a couple of tags. Experimental results reveal that iEKF algorithm offers more accuracy than the LSE method, e.g., by calculating the coordinates of the head tag, the mean deviation of iEKF is 7.16 cm, lower than that of LSE (11.70 cm). In the improved extend Kalman filter, the influence of parameter *N* for window function has also been discussed, and the suggestion of reasonable range of *N* is given. 

It has to be pointed out that there are also some disadvantages in our solution. We can achieve good posture recognition performance, but many tags are needed to keep working, and their energy is an important concern. In future work, we will transplant the algorithm to the processor, and focus on the further improvement of wearable technologies coupled with different kinds of postures, such as walking, sitting, sleeping, and so on. The test of the robustness and stability of the system also need to be carried out. We believe the prospect of applications for elderly care is vast.

## Figures and Tables

**Figure 1 sensors-19-01464-f001:**
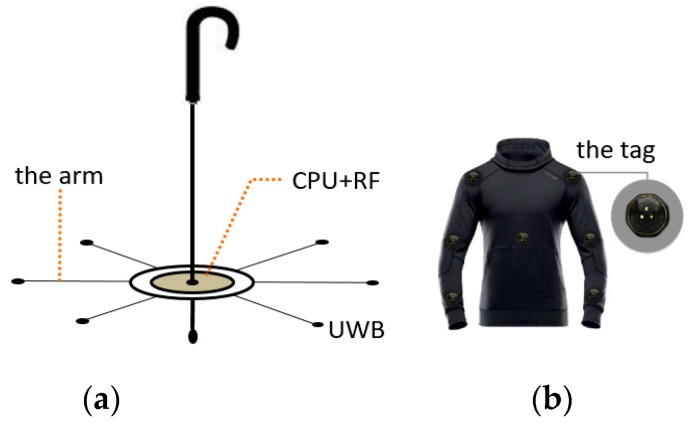
Positioning umbrella and receiving tags. (**a**) Positioning umbrella. (**b**) Receiving tags attached on clothes.

**Figure 2 sensors-19-01464-f002:**
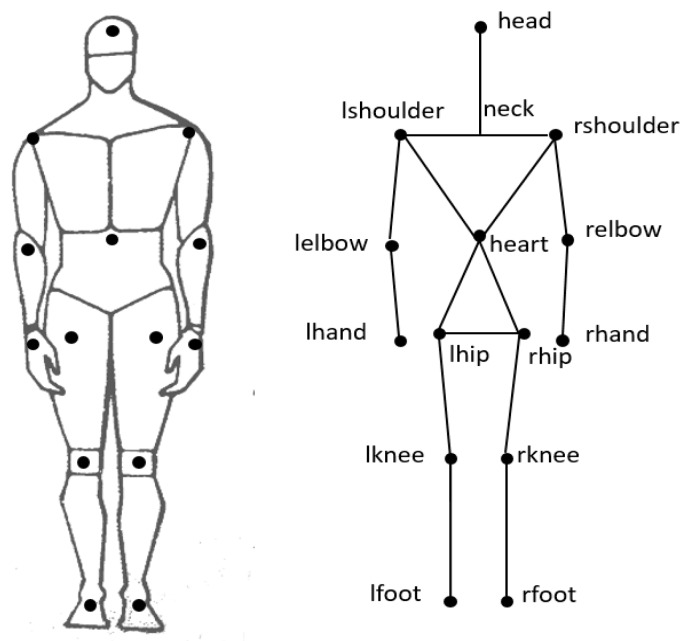
Distribution of UWB receiving tags.

**Figure 3 sensors-19-01464-f003:**
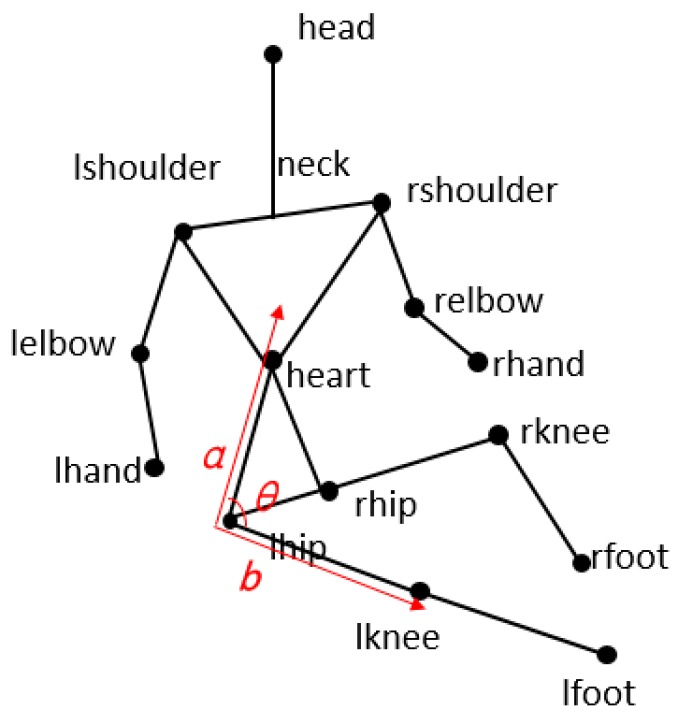
Posture angles of falling down.

**Figure 4 sensors-19-01464-f004:**
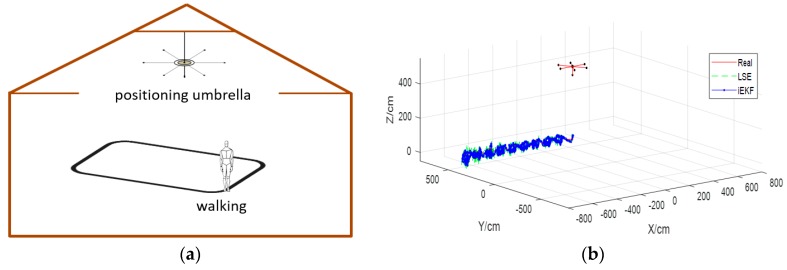
Left foot trajectory in walking model. (**a**) Walking model. (**b**) Trajectory of left foot.

**Figure 5 sensors-19-01464-f005:**
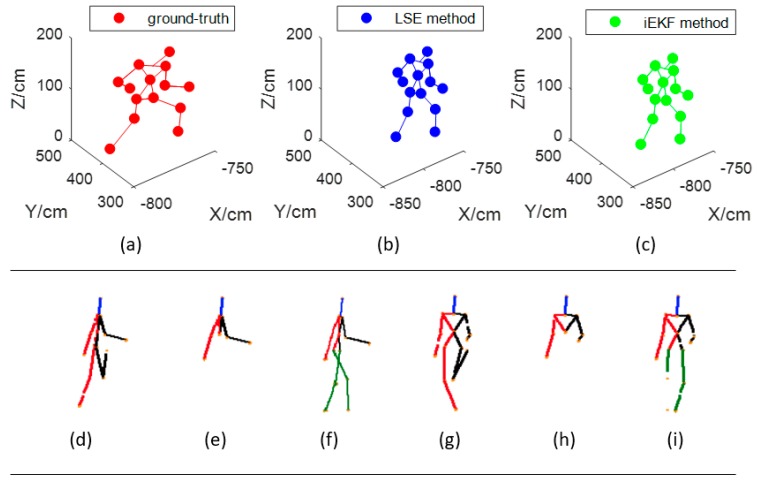
Detecting result for 14 key joints in walking model. (**a**), (**b**) and (**c**) are the first group detected by our designed indoor position system. (**a**) is the original data generated by UE 4.0, (**b**) is the 14 key joints detected by LSE method, and (**c**) is the result detected by iEKF. Figures from (**d**) to (**i**) are the results detected by Kinect sensor. (**d**) and (**g**) are the expected results. (**e**) and (**h**) miss some joints due to shelter or failed detection. (**f**) and (**i**) are the repaired results in literature [30].

**Figure 6 sensors-19-01464-f006:**
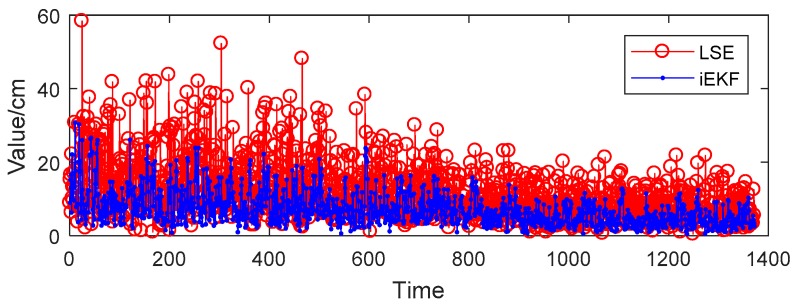
Deviation performances of the two algorithms for head tag.

**Figure 7 sensors-19-01464-f007:**
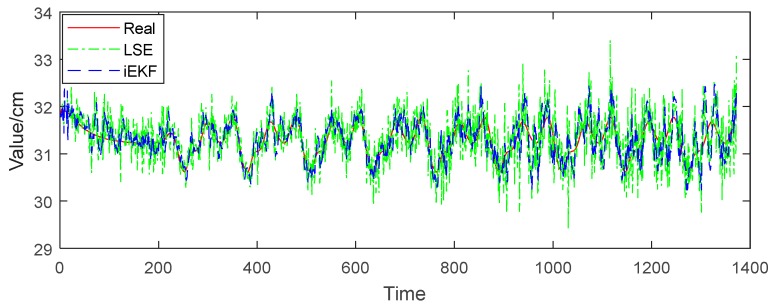
Norm comparison of vector $I_{\mathrm{heart}\text{-}\mathrm{to}\text{-}\mathrm{lshoulder}}$.

**Figure 8 sensors-19-01464-f008:**
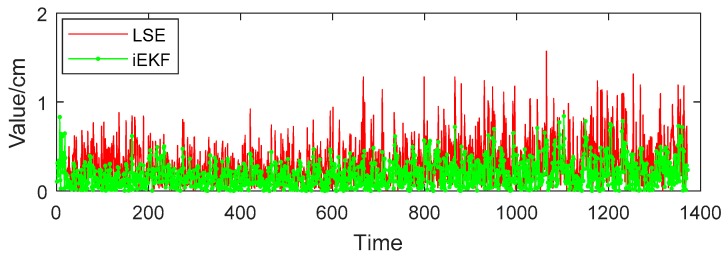
Deviation comparison of vector $I_{\mathrm{heart}\text{-}\mathrm{to}\text{-}\mathrm{lshoulder}}$.

**Figure 9 sensors-19-01464-f009:**
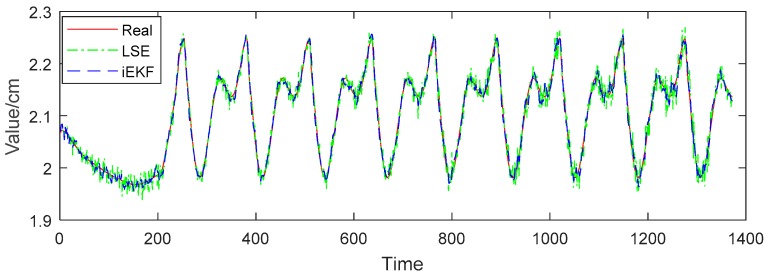
Comparison of two algorithms for $\theta_{\mathrm{heart}\text{-}\mathrm{lshoulder}\text{-}\mathrm{lelbow}}$.

**Figure 10 sensors-19-01464-f010:**
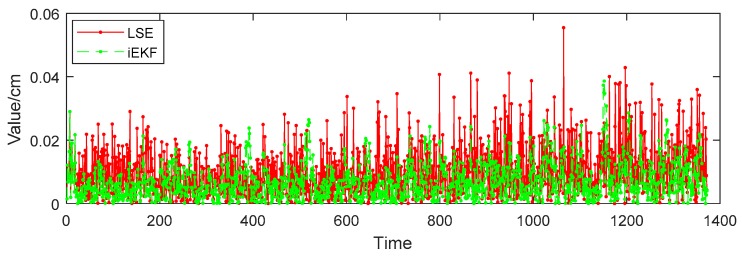
Deviation comparison for angle $\theta_{\mathrm{heart}\text{-}\mathrm{lshoulder}\text{-}\mathrm{lelbow}}$.

**Figure 11 sensors-19-01464-f011:**
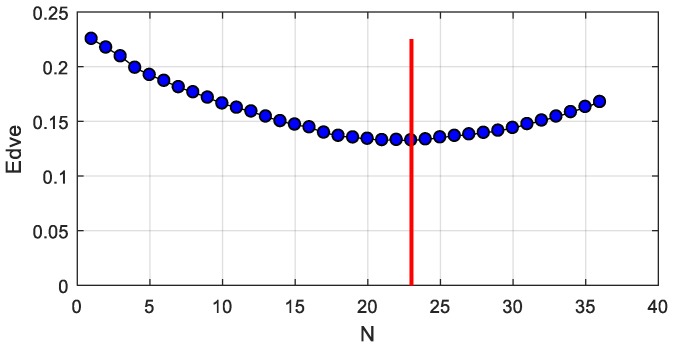
Influence of N for mean deviation of vector $I_{\mathrm{lshoulder}\text{-}\mathrm{to}\text{-}\mathrm{lelbow}}$.

**Table 1 sensors-19-01464-t001:** Key joints of human body.

Position	Tag Name	Position	Tag Name
Head	head	Central	heart
Left Shoulder	lshoulder	Right Shoulder	rshoulder
Left Elbow	lelbow	Right Elbow	relbow
Left Hand	lhand	Right Hand	rhand
Left Hip	lhip	Right Hip	rhip
Left Knee	lknee	Right Knee	rknee
Left Foot	lfoot	Right Foot	rfoot

**Table 2 sensors-19-01464-t002:** Structure vectors of human body.

Vector Name	Position	Vector Name	Position
*I _lelbow-to-lhand_*	Left Hand	*I* _relbow-to-rhand_	Right Hand
*I _lshoulder-to-lelbow_*	Left Arm	*I _rshoulder-to-relbow_*	Left Arm
*I _lhip-to-lknee_*	Left Leg	*I _rhip-to-rknee_*	Left Leg
*I _lknee-to-lfoot_*	Left Foot	*I _rknee-to-rfoot_*	Left Foot
*I _head-to-lshoulder_*	Head-Left Shoulder	*I _head-to-rshoulder_*	Head-Right Shoulder

**Table 3 sensors-19-01464-t003:** Posture angles of human body.

Posture Angle	Position	Posture Angle	Position
*θ _head-heart-lshoulder_*	Head-Heart-Left Shoulder	*θ _head-heart-rshoulder_*	Head-Heart-Right Shoulder
*θ _heart-lshoulder-lelbow_*	Heart-Left Shoulder-Elbow	*θ _heart-rshoulder-relbow_*	Heart-Right Shoulder-Elbow
*θ _lshoulder-lelbow-lhand_*	Left Shoulder-Elbow-Hand	*θ _rshoulder-relbow-rhand_*	Right Shoulder-Elbow-Hand
*θ _lhip-lknee-lfoot_*	Left Hip-Knee-foot	*θ _rhip-rknee-rfoot_*	Right Hip-Knee-foot

**Table 4 sensors-19-01464-t004:** Mean deviation comparison in tag position, vector norm and posture angle.

Name	LSE	iEKF
head	11.70 cm	7.16 cm
left shoulder	11.83 cm	7.24 cm
*I _heart-to-lshoulder_*	0.32 cm	0.20 cm
*I _lshoulder-to-lelbow_*	0.22 cm	0.14 cm
*θ _head-heart-lshoulder_*	0.0075 rad	0.0046 rad
*θ _heart-lshoulder-lelbow_*	0.0104 rad	0.0073 rad

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
