# Peer review of "A Posture Recognition Method Based on Indoor Positioning Technology"

_sensors, 2019, doi:10.3390/s19061464_

Round 1

Reviewer 1 Report

Some abbreviations are presented before their explanation, e.g., UWB in abstract.

Some relevant research studies about the use of inertial sensors (acceleration sensors are not presented).

The conclusions should be improved.

The results should be compared with other systems and/or other methods with the same dataset.

Is the dataset publicly available?

As it use human data, are you taking in account the new GDPR?

The language should be improved.

Can you describe future work? How useful is your research. The motivation is not presented in the paper.

The number and quality of the references should be improved. It is a study widely researched. You need verify the related work.

As example the use of data fusion should be explained.

The study design and experiments should be described in a separate section than the results.

As example:

I. M. Pires, N. M. Garcia, N. Pombo, F. Flrez-Revuelta, S. Spinsante, and M. C. Teixeira, Identification of Activities of Daily Living
through Data Fusion on Motion and Magnetic Sensors embedded on Mobile Devices, Pervasive and Mobile Computing, vol. 47, pp. 78-93, 2018. doi: 10.1016/j.pmcj.2018.05.005

I. M. Pires, N. M. Garcia, N. Pombo, and F. Flrez-Revuelta, Identification of Activities of Daily Living Using Sensors Available in offthe-shelf Mobile Devices: Research and Hypothesis, Ambient Intelligence-Software and Applications-7th International Symposium on Ambient Intelligence (ISAmI 2016), 2016, pp. 121-130.

Lee, Seon-Woo, and Kenji Mase. "Activity and location recognition using wearable sensors." IEEE pervasive computing 1.3 (2002): 24-32.

Lyons, Damian M. "System and method for permitting three-dimensional navigation through a virtual reality environment using camera-based gesture inputs." U.S. Patent No. 6,181,343. 30 Jan. 2001.

I. Pires, N. Garcia, N. Pombo, and F. Flrez-Revuelta, From Data Acquisition to Data Fusion: A Comprehensive Review and a Roadmap for the Identification of Activities of Daily Living Using Mobile Devices, Sensors, vol. 16, p. 184, 2016.

Rafii, Abbas, et al. "Gesture recognition system using depth perceptive sensors." U.S. Patent No. 9,959,463. 1 May 2018.

Valin, Myriam, et al. "MEMS-based method and system for tracking a femoral frame of reference." U.S. Patent No. 9,901,405. 27 Feb. 2018.

I. M. Pires, N. M. Garcia, and F. Flrez-Revuelta, Multi-sensor data fusion techniques for the identification of activities of daily living using mobile devices, Proceedings of the ECMLPKDD 2015 Doctoral Consortium, European Conference on Machine Learning and Principles and Practice of Knowledge Discovery in Databases, Porto, Portugal, 2015.

Author Response

Dear Editors and Reviewers:

  Thank you for your letter and for your comments concerning our manuscript entitled “A Posture Recognition Method Base on Indoor Positioning Technology”. Manuscript id is sensors-432579. Those comments are all valuable and very helpful for revising and improving our paper, as well as the important guiding significance to our researches. We have studied comments carefully and have made correction which we hope meet with approval. The main corrections in the paper and the responds to your comments are in the attached word document.

Reviewer 2 Report

Summary:
This paper presents a posture recognition method based on wearable tags on human body. It uses least square method and extended Kalman filter to achieve accurate recognition and positioning.
Due to the missing results and findings, my recommendation for this paper is a major revision.

Comments:
The English writing requires much improvement. There are many grammatical mistakes in the text (e.g. abstract). To name a few:
- L17: It is presented in the paper that a new [...] scheme.
- L25: it can derived
- L38: can mainly divided
- L67: it is proposed a new ... in the paper. --> This paper proposes a new... OR A new ... is proposed in this paper.

Jargons are being used before full introduction: e.g. UWB in abstract, TODA

Where does this work stand with respect to the existing work and state-of-the-art? There is no comparison with state-of-the-art whatsoever. As authors mentioned, the posture recognition is a hot topic with growing research attention. There must be similar attempts to solve the same problem. What are the (dis-)advantages of proposed work compared to the existing ones?

Authors mentioned that they implemented the algorithm on an STM32 microprocessor. However, the experimental results lack the details of evaluation on the hardware platform. I recommend the authors to add a subsection to the article, provide technical details of this microprocessor (cpu frequency, memory, ...), and report the performance of the algorithm on the platform. They can report the resource usage on the platform including execution time, code size, memory (if possible).

The provided results are not picturing how well the algorithm performs in "posture recognition". They mainly include the deviation of angles. How these parameters can be translated to a more meaningful metric for the user? In other words, authors should clarify what is the final output of the application, and how close their calculation became to the golden output.

Author Response

(The authors gave the same response as above.)

Round 2

Reviewer 1 Report

The conclusions may be improved and extended.

Author Response

Dear editors and reviewers,

     Thank you for your letter and for the comments concerning our manuscript entitled “A Posture Recognition Method Base on Indoor Positioning Technology”. Manuscript id is sensors-432579. Your comments are all valuable and very helpful for revising and improving our paper. We have made correction, in the attachment, which we hope meet with approval.

Comments and Suggestions for Authors:

The conclusions may be improved and extended.

We really appreciate your suggestion, the revision edition has been extended. A summary of the introduction has been added. The algorithms performance and experiment result has been discussed, and the disadvantage of the system and future improvement are presented.

Yours sincerely,

Xiaoping Huang

Reviewer 2 Report

The authors only addressed the minor comments (editorial mistakes), and the major concerns regarding the experimental results still remain. The provided arguments are convincing. Therefore, my recommendation is major revision.

The article still lacks some results. For instance, comparison with state-of-the-art is still missing. Qualitative comparison with a wide category of posture recognition methods (i.e. computer vision) which is technically far from this work cannot replace the real comparison with closer existing techniques.

Author Response

Dear Reviewer:

Thank you for your letter and for the comments concerning our manuscript entitled “A Posture Recognition Method Base on Indoor Positioning Technology”. Manuscript id is sensors-432579.Your comments are all valuable and very helpful for revising and improving our paper. We have made correction, in the attachment, which we hope meet with approval.

Round 3

Reviewer 2 Report

I am satisfied with the changes made in the revised manuscript.